

# Choosing mates based on the diet of your ancestors: replication of non-genetic assortative mating in *Drosophila melanogaster*

Michael A. Najarro, Matt Sumethasorn, Alexandra Lamoureux and Thomas L. Turner

Ecology, Evolution, and Marine Biology Department, University of California, Santa Barbara, USA

## ABSTRACT

Assortative mating has been a focus of considerable research because of its potential to influence biodiversity at many scales. *Sharon et al. (2010)* discovered that an inbred strain of *Drosophila melanogaster* mated assortatively based on the diet of previous generations, leading to initial reproductive isolation without genetic evolution. This behavior was reproduced by manipulating the microbiome independently of the diet, pointing to extracellular bacterial symbionts as the assortative mating cue. To further investigate the biological significance of this result, we attempted to reproduce this phenomenon in an independent laboratory using different genotypes and additional mating assays. Supporting the previous result, we found that a different inbred strain also mated assortatively based on the diets of previous generations. However, we were unable to generate assortative mating in an outbred strain from North Carolina. Our results support the potential for non-genetic mechanisms to influence reproductive isolation, but additional work is needed to investigate the importance of this mechanism in natural populations of *Drosophila*.

## INTRODUCTION

Assortative mating is a term that describes when individuals choose mates on the basis of a shared phenotype or genotype. Assortative mating (sometimes referred to as positive assortative mating) has been found in many natural populations, where it can have important effects on diversity (*Wright, 1921*; *Jiang, Bolnick & Kirkpatrick, 2013*). Assortative mating is also key to behavioral isolation between populations, which is a crucial step in the speciation process (*Coyne & Orr, 2004*).

The ease with which behavioral isolation can arise is an important variable in some models of sympatric speciation (*Kondrashov & Kondrashov, 1999*), with non-genetic causes of assortative mating receiving increasing attention as important factors (*Pfennig & Servedio, 2013*). Assortative mating is known to be affected by non-genetic inheritance in some systems, such as cultural transmission in birds and fish (*Crews et al., 2007*;

Corresponding author
Michael A. Najarro,
michael.najarro@lifesci.ucsb.edu

*Pfennig & Servedio, 2013*. Bacterial symbionts in insects have also been shown to influence both pre-zygotic reproductive isolation (by influencing mate choice) and post-zygotic reproductive isolation (by reducing the fitness of hybrids) in several species, including *Drosophila* (*Bordenstein, O'Hara & Werren, 2001*; *Koukou et al., 2006*; *Miller, Ehrman & Schneider, 2010*; *Brucker & Bordenstein, 2012*). Until recently, these cases were all thought to be due to intracellular symbionts like *Wolbachia* rather than extracellular symbionts like gut bacteria. Gut bacteria, however, were recently shown to influence the viability of hybrids in *Nasonia* (*Brucker & Bordenstein, 2013*; *Chandler & Turelli, 2014*), and several studies of symbiont-mediated assortative mating used antibiotics that likely perturbed extracellular symbionts as well as intracellular ones (*Koukou et al., 2006*; *Miller, Ehrman & Schneider, 2010*).

*Sharon et al. (2010)* recently demonstrated that extracellular bacterial symbionts, likely living in the gut, can be vertically transmitted in laboratory *Drosophila*, and that flies mate assortatively based on these symbionts (at least in the lab, in the particular inbred strain studied). In this paradigm, populations of an inbred line (Oregon-R) were reared for one or more generations on different diets, and were then reared for a single generation on a common diet. The flies reared on a common diet mated assortatively based on the diets of their ancestors. Assortative mating was significant after only two generations on different diets, making it unlikely that this phenomenon was a byproduct of adaptation to diet. Instead, the authors found that one diet (starch) induced major changes in the microbiome of the flies, and that these changes persisted after the original diet was restored. Essentially, in this particular experimental setup, the microbiome of the flies was inherited non-genetically. Further experiments showed that assortative mating could be reproduced by directly manipulating the microbiome alone (*Sharon et al., 2010*). It is possible that the previous work of Dodd, who found assortative mating between *D. pseudoobscura* populations reared on starch and maltose diets, was also due to heritable microbes rather than adaptation, as was assumed at the time (*Dodd, 1989*). Many other experiments have reared *Drosophila* species in different environments for dozens to hundreds of generations and some have found assortative mating after this time; it is possible that some of these cases are also non-genetic (*Coyne & Orr, 2004*, pp. 88–89).

This is the only case we are aware of that clearly implicates extracellular microbes in behavioral isolation. This work is, therefore, an important component of our emerging understanding of the role the microbiome plays in assortative mating and speciation. However, because some published results are likely false, replication of these results would be valuable (*Ioannidis, 2005*). Moreover, the original results were partially confounded by pseudo-replication, and though the authors suggested that changes in pheromone composition could be responsible, we feel that the mechanism behind this phenomenon is not yet established. We therefore attempted to generate assortative mating in additional genotypes, both inbred and outbred, and used an additional type of mating assay to explore the possible mechanism. We were able to generate assortative mating based on the diets of previous generations, in support of published results. However, because we

were only able to generate assortative mating in an inbred strain, the relevance of this phenomenon in natural populations remains to be determined.

## MATERIALS AND METHODS

### Drosophila rearing

Two genotypes were used in these experiments. The first was an inbred strain, Canton-S, acquired from the Bloomington Drosophila Stock Center. The second strain was an outbred strain created by mass-mating 173 wild isofemale lines collected in Raleigh, North Carolina by Trudy Mackay (*Mackay et al., 2012*). We refer to this strain as allRAL, following previous publications using the same stock; these strains were also acquired from the Bloomington Stock Center (*Turner, Miller & Cochrane, 2013*; *Pischedda et al., 2014*). This population has been maintained in the lab for 5 years since the original lines were mixed, at a population size of a few hundred individuals of each sex per generation. This strain is therefore likely to have much more genetic diversity than lab strains that were intentionally inbred and then reared in the lab for decades, like Canton-S and Oregon-R. We refer to this strain as an outbred strain, though it should be noted that it is likely more inbred than natural populations of *D. melanogaster*.

All flies were maintained in 35 mm vials with a 12 h light-dark cycle at 25 °C. Each genotype was split into 2 populations to be reared on each diet, with each of these populations maintained in a set of twenty 35 mm vials. Some populations continued to be reared on a standard cornmeal, molasses, yeast diet (CMY) while the others received starch media, which was a modified CMY recipe that did not contain molasses and substituted potato starch for cornmeal (90% water, 1.5% agar, 5% yeast, 0.5% propionic acid, 3% starch (*Sharon et al., 2010*)). Fresh starch media was cooked in a small batch on a hot plate at the end of every week and poured into vials. The vials were covered and allowed to dry overnight and were then refrigerated. The food was used the following week. CMY media was cooked on a larger scale every two weeks, allowed to dry, and then refrigerated. Following *Sharon et al. (2010)*, flies destined for starch diets were first reared for a generation on a transitional diet that was a 50/50 mix of the two diets. We found that excluding this step resulted in very high mortality on the starch diet. The entire experimental cycle of the starch-reared flies therefore consisted of rearing the flies on a transitional diet (step 1), then a starch diet for at least two generations (step 2), then finally back to a CMY diet (step 3); flies in the CMY treatment were always reared on CMY. Flies from step 3 were used in behavioral experiments, so these flies had all been reared in the same type of food for one generation to standardize non-heritable effects of the environment. This does not mean they were reared in a common environment, however, as they were reared in separate vials and potentially had different microbial flora or other vertically transmitted environmental differences. Following *Sharon et al. (2010)*, flies were maintained in step 2 (starch food) for variable lengths of time. Populations were initially established in January 2013 and experiments were tested in March 2013; these flies were maintained on starch for only 2 generations. A second set of populations was established in June of 2013 and maintained on starch for 20 generations before offspring were collected

and moved to CMY food for behavioral experiments. For all experiments, starch and CMY treatments were set up for our inbred Canton-S strain and for our outbred allRAL strain. These two strains were assayed independently throughout the experiments.

## Virgin fly collection and wing clipping

Virgin flies were collected from the final CMY generation by collecting flies that were less than 6 h post-eclosion, 11 days after oviposition. Virgin flies from 20 different vials of the same treatment were pooled in each block, anesthetized with carbon dioxide, and then separated by sex into groups of 10 or 20 on fresh CMY media. On the 4th day post-collection, all flies were re-anesthetized for wing clipping. *Sharon et al. (2010)* clipped the wings of flies from one diet treatment only, and used these marks to determine mating pattern. Past research has found no significant effects of wing clipping on male courtship or mating success (e.g., *Van den Berg et al., 1984*), but we still felt it would be better to clip the wings of all flies to control for any effect of wing clipping. We therefore clipped one treatment on the right side and the other on the left side. The posterior portion of one wing of each virgin fly was ablated to form a blunted wing. Both sexes within a diet treatment received a wing clip to the wing on the same side while the other diet treatment received a wing clip on the opposite wing. Wing clipping patterns were alternated across replicate experiments to randomize any effect of courtship bias for a particular side. Flies were allowed to recover from anesthetization for 18–20 h.

## Mate choice trials

Mate choice experiments were performed when flies were five days post-eclosion, 18 to 20 h after wing clipping. Mating arenas were 3 ml wells of 24-well tissue culture plates (Falcon). We used a four-fly mate choice design to replicate the methods of *Sharon et al. (2010)*. First, a male from each diet treatment (starch or CMY) was loaded sequentially into each well. Males from one treatment were loaded until all wells were filled, followed by males from the second treatment. The loading order of each male with respect to treatment was alternated between plates. The females from both diet treatments were loaded into each well within seconds of each other to prevent one female from receiving more courtship than the other; the loading order of females with respect to diet was also alternated across plates. Flies were loaded by mouth aspirator; the tips of the aspirator that are in contact with the flies were changed depending on the rearing media and strain of the fly being loaded. Plates were observed for one hour from the time the plates were completely filled. We recorded the wing-clipping pattern of only the first mating pair within a replicate using a hand lens, as a second copulation within this well would not be an independent observation. Note that both copulations were erroneously considered independent in the original *Sharon et al. (2010)* publication, but that a correction was later published (*Sharon et al., 2013*).

We also performed a different mate choice experiment, which we refer to as three-fly mate choice. In these trials, only a single fly was loaded for one sex (male or female) and designated a "chooser"; flies of the opposite sex from each treatment were loaded as candidate mates. Based on the sex of the chooser, the three-fly mate choice experiment
can more specifically be called "female choice" or "male choice" experiments. We use this terminology due to convenience, and following previous work, but note that it is likely naïve to assume that preferences of the mating candidates do not also play a role in these experiments. The chooser was loaded into arenas first, followed by both candidate mates within seconds of each other.

## Data analysis

Several covariates existed within the four-fly mate choice experiment (date of assay, time of assay, number of males held per vial before assay, and wing clipping effect). Wing clipping effect was a covariate that controlled for "handedness" in each sex with regard to the use of a wing in courtship. In other words, we controlled for males preferring to court with one particular wing, and females preferring to receive courtship from one wing of a male. The significance of each covariate was assessed using a logistic regression model. No covariates were significant ($p > 0.05$), so data were analyzed together. Binomial tests were then used to determine if mating was significantly different from random with respect to treatment for this and the 3-fly mate choice trials. These tests were done in R using the binom.test() function, which also reports confidence intervals estimated using the procedure of *Clopper & Pearson (1934)*. With permission from the previous authors, the frequencies of assortative and disassortative mating from the four-fly mate choice experiments were compared to raw data provided by *Sharon et al. (2010)* to determine if the data collected supported original findings.

## RESULTS

In support of previous findings, we found that the diet of ancestors affected the mate choice of later generations, but only in the inbred Canton-S strain (Fig. 1 and Table 1). In the four-fly experiment, 58% of copulations were assortative in Canton-S, where 50% are expected under random mating (binomial $p = 0.004$, 95% confidence interval $= 0.52$–$0.63$, $N = 344$). The outbred allRAL strain, in contrast, did not show any significant deviation from random mating, with 52% of copulations assortative (binomial $p = 0.53$, 95% confidence interval $= 0.46$–$0.58$, $N = 302$). We used raw data provided by *Sharon et al. (2010)* to calculate that the Oregon-R inbred strain mated assortatively 61% of the time (binomial, $p = 1.08E–5$, 95% confidence interval $= 0.56$–$0.66$, $N = 385$), similar to our data for Canton-S. When Canton-S mated assortatively, we saw no significant difference in which flies mated first: starch flies were observed to mate first 52% of the time, and CMY flies the other 48% of the time (binomial $p = 0.56$). These data are consistent with the published data using Oregon-R, where starch flies were the first to mate 46% of the time (binomial $p = 0.27$).

We gathered additional information regarding mate choice using a three-fly mate choice design, where a single fly of one sex is presented with a choice of flies of the other sex (in contrast to the four-fly mate choice design presented above and in *Sharon et al. (2010)*, which has two flies of both sexes). We tested all four possible combinations of flies: a starch female with both males, a CMY female with both males, a starch male with both females, and a CMY male with both females. Sample sizes were smaller than the four-fly experiment
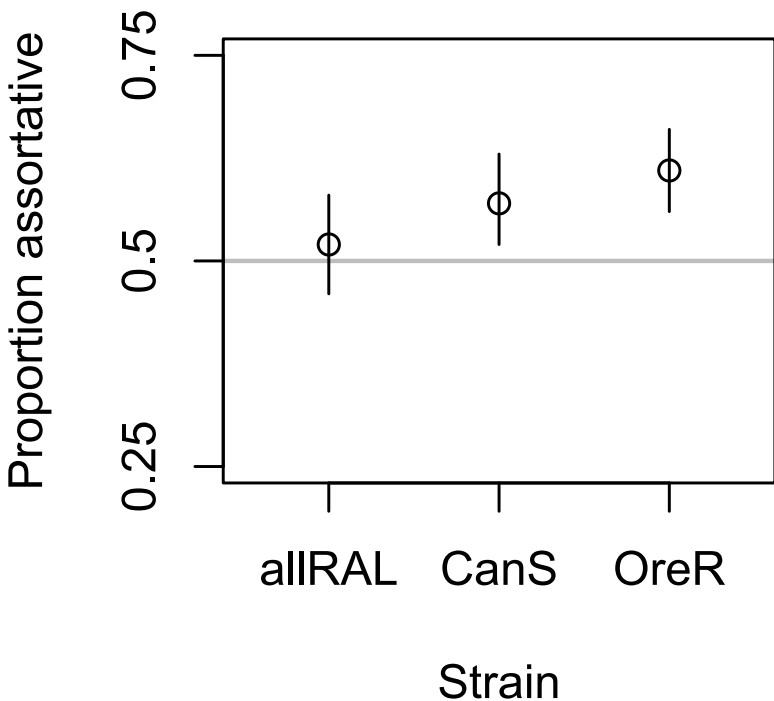

**Figure 1** **Mate choice behavior among strains.** The frequency of assortative mating for each strain tested is shown, with Oregon-R data from *Sharon et al. (2010)* presented for comparison. Vertical lines are 95% confidence intervals; sample sizes were 302, 344, and 385 for allRAL, Canton-S, and Oregon-R, respectively.

**Table 1** **Results of four-fly mate choice experiments, wherein a male and female from each treatment are mixed.**

|  | Canton-S | allRAL | Oregon-R |
|---|---|---|---|
| $N$ | 344 | 302 | 385 |
| Assortative | 199 | 157 | 236 |
| Disassortative | 145 | 145 | 149 |
| Starch × Starch[*] | 100 | 83 | 109 |
| CMY × CMY[*] | 91 | 67 | 127 |
| CMY × Starch[*] | 67 | 72 | 81 |
| Starch × CMY[*] | 74 | 68 | 68 |
| Proportion assortative | 0.57 | 0.52 | 0.61 |
| $P$ | 0.004 | 0.53 | 1.1 E 10–5 |
| Confidence interval | 0.52–0.63 | 0.46–0.58 | 0.56–0.66 |

**Notes.**
[*] The treatments of the first pair are listed with female first. These rows denote those cases where one pair of flies clearly mated first.

due to the need to run these multiple comparisons; to maximize sample size, we only investigated Canton-S because of the positive results of the four-fly design (Table 2).

Interestingly, in both scenarios where the "chooser" fly was from the CMY treatment, the proportion of copulations that were assortative were nearly identical to the 4-fly

**Table 2  Results of three-fly mate choice experiments, wherein a single fly of one sex (the chooser) is presented a fly of the opposite sex from each treatment.**

| Choosing fly | Assortative | Disassortative | N | Proportion assortative | P | Confidence interval |
|---|---|---|---|---|---|---|
| Starch female | 89 | 107 | 196 | 0.454 | 0.23 | 0.38–0.53 |
| CMY female | 113 | 88 | 201 | 0.562 | 0.09 | 0.49–0.63 |
| CMY male | 106 | 79 | 185 | 0.572 | 0.06 | 0.50–0.65 |
| Starch male | 88 | 78 | 166 | 0.530 | 0.49 | 0.45–0.61 |

experiment, in which 58% of copulations were assortative (Table 2). When a female CMY fly was combined with both males, 56% of copulations were assortative, and when a male CMY fly was combined with both females, 57% of copulations were assortative. These results were only marginally significant due to decreased sample sizes (binomial $p = 0.09$, $N = 201$ and $p = 0.06$, $N = 185$, for females and males, respectively). If we use Fisher's combined probability test (i.e., Fisher's method) to consider these data together, copulations are significantly assortative when CMY flies are choosers ($p = 0.03$). In contrast, the cases where starch flies were the "chooser" were not significant (Table 2).

## DISCUSSION

In *2010*, Gil Sharon and colleagues published a report describing assortative mating based on gut bacteria in *Drosophila melanogaster*. We found these data surprising and interesting, and have attempted to expand on the understanding of this phenomenon here. In partial replication of this previous result, we found that flies from the Canton-S strain mate assortatively based on the diets of their ancestors. Though we did not explicitly investigate the presence of any particular bacterial species, this is consistent with the pattern observed by *Sharon et al. (2010)* with the Oregon-R strain. In contrast, we saw random mating in flies from our outbred allRAL strain. With only two inbred and one outbred strain tested thus far, it is not possible to determine if this was because the flies were outbred or because of a genotype-by-environment interaction. It is also possible that the initial bacterial community was different in this strain for stochastic reasons, and that this community did not produce the affects seen in the other strains. In any case, when the three different genotypes are considered together, the data are not consistent with random mating (Fisher's combined probability test, $p = 4.03E–6$).

Our data make it less likely that the pattern seen by *Sharon et al. (2010)* was a statistical artifact, or that it was a phenomenon particular to a single lab strain. Moreover, we were able to reproduce assortative mating in a different mate-choice assay. When either a single male or female CMY fly was presented with both types of the opposite sex, they mated assortatively more often than expected by chance. This was not seen when starch flies were the "choosers", supporting the hypothesis that CMY flies are discriminating against starch flies when given a choice, rather than vice versa. This is interesting with respect to the nature of the two diet treatments. The starch diet is stressful for *D. melanogaster*, as evidenced by the need to rear a generation on a transitional 50% starch food diet before rearing on

starch only in order to avoid nearly complete mortality in inbred strains (see methods). In contrast, CMY food is a standard *Drosophila* rearing media. Though the flies used in our experiments were all reared on CMY media for a single generation, the results of *Sharon et al. (2010)* suggest that those with starch-reared ancestors have lost most of the diversity in their microbiome; flies could therefore still be stressed after one generation on CMY food. This leads us to hypothesize that the pattern seen here could simply be due to assortative mating between the healthiest males and females. For example, both male and female *D. melanogaster* prefer large mates to small mates (*Partridge & Farquhar, 1983*; *Partridge, Hoffmann & Jones, 1987*; *Pitnick, 1991*; *Byrne & Rice, 2006*; *Long et al., 2009*). If flies reared on starch lose important bacterial symbionts, then their descendants might be smaller, less vigorous, or otherwise less desirable. We further speculate that this pattern might not be seen in our outbred allRAL stock due to the increase in vigor in outbred vs. inbred strains. If outbred individuals reared on starch are able to either maintain their bacterial community or deal better with its reduction, then assortative mating might not be seen. In this scenario, we might predict that mating would be significantly *disassortative* when single starch flies are given an option; mating was only disassortative in the case of starch females, and not significantly so (45% of copulations were assortative, 95% CI [38–53]). Further investigating these hypotheses might be a productive direction for future research.

Our proposed mechanism (assortative mating based on quality or condition) is very straightforward but could play an important role in speciation in nature. If some individuals in a population use a sub-optimal habitat, they might mate assortatively simply due to rejection by other individuals using a more typical habitat. If these individuals mate assortatively by default, rare recessive alleles would be more likely to become homozygous, and alleles at unlinked loci could establish gametic disequilibrium, both of which could promote local adaptation. Of course, once these more-fit genotypes emerged, they would no longer be discriminated against by other individuals on typical habitat, so a genetic assimilation-like mechanism would be required to continue reproductive isolation (*Waddington, 1953*; *Badyaev, 2005*). Needless to say, these ideas are speculative and require further investigation via theoretical models and experiments in the field.

In addition to the caveats already provided, we feel it is important to emphasize that the experimental setup used in our work and the work of *Sharon et al. (2010)* should only be considered a proof-of-concept because the particular diets used are quite artificial. Furthermore, the degree to which gut bacteria are vertically transmitted in nature remains unknown. Extracellular bacteria might be more reliably transmitted to offspring in the lab than the field, because flies with different ancestral treatments were reared in separate vials in our experiment. In contrast, a patch in nature could be used by flies that developed on many different types of patches, such that their combined bacterial communities could seed the substrate for the communal pool of offspring. Nonetheless, the results obtained from our experiments demonstrate that under some conditions, changes in the diet of ancestral generations can promote assortative mating in later generations. Our data support the findings of *Sharon et al. (2010)*, and point to assortative mating based on

general vigor as a possible mechanism. We hope that these results will motivate continued work in this area.

## ACKNOWLEDGEMENTS

We would like to thank Leslie Benitez, Melina Allahverdian, and Erika Kim for helping collect data and Alison Pischedda, Michael Shahandeh, Steven Proulx, and William Rice for thoughtful discussion regarding experimental design, analysis, and interpretation. This work would not have been possible without the community supported resources available at the Bloomington Drosophila Stock Center.

### Funding

Funding was provided by the University of California Santa Barbara and the National Institutes of Health (R01 GM098614). The funders had no role in study design, data collection and analysis, decision to publish, or preparation of the manuscript.

### Grant Disclosures

The following grant information was disclosed by the authors:
University of California Santa Barbara and the National Institutes of Health: R01 GM098614.

### Competing Interests

The authors declare there are no competing interests.

### Author Contributions

- Michael A. Najarro conceived and designed the experiments, performed the experiments, analyzed the data, wrote the paper, prepared figures and/or tables, and reviewed drafts of the paper.
- Matt Sumethasorn and Alexandra Lamoureux performed the experiments, and reviewed drafts of the paper.
- Thomas L. Turner conceived and designed the experiments, analyzed the data, wrote the paper, prepared figures and/or tables, and reviewed drafts of the paper.

### Supplemental Information

Supplemental information for this article can be found online at http://dx.doi.org/10.7717/peerj.1173#supplemental-information.

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
