# Peer review of "Choosing mates based on the diet of your ancestors: replication of non-genetic assortative mating in Drosophila melanogaster"

_PeerJ, doi:10.7717/peerj.1173_

## Round 0.1 · original submission · Minor Revisions

The three reviewers concur that the research findings presented in this manuscript are robust, and include mostly minor comments seeking clarify in some parts. I agree and recommend this manuscript for publication pending some minor revisions as suggested by reviewers.

One reviewer requests that additional data be presented in the main body of the manuscript (rather than being reported as un-summarized data in the supplemental materials), and I concur. This reviewer also suggests additional microbiome experiments. These would be nice, but I won’t consider this a pre-requisite for publication.

I am curious about the use of the term “epigenetic” in the title. Used in this sense, are you considering “epigenetic” every type of across-generation transmission of information that is not genetic? That is, would a maternal provisioning effect be considered “epigenetic”? Some consider epigenetics specifically to refer to molecular alterations of the epigenome – especially in more modern parlance as mentioned also by one reviewer. So it might be worth qualifying your use of the term. Reviewer 3 raises some similar concerns, and their suggestion of some modified text and terminology is warranted.

Additionally, since starch and CMY lineages are separated by only one generation, can you rule out the effect of direct exposure (which would rule out anyone’s use of the term “epigenetic”). That is, is it possible that gametes were directly exposed to altered diet, such that when those gametes combine to form the next generation that direct exposure becomes manifest as altered “stress” phenotype. If this is possible, then to demonstrate trans-generational inheritance would require extending the experiment out another generation of both lineages back on CMY diet.

I’m also struck of course by the difference of effect between the inbred and outbred strains. Mike Skinner’s group also studies epigenetic inheritance – mostly in the context of chemical toxicant effects that are propagated across generations in rodent models. So perhaps lots of important differences compared to your studies. Curiously, his group also finds that propagation of information across generations is different between inbred and outbred strains, but in the opposite direction of your findings. E.g., see http://www.ncbi.nlm.nih.gov/pubmed/23041264 - wherein they find that phenotype is more strongly heritable in the OUTBRED group, but weaker in the inbred group.

·

Basic reporting

The manuscript is clearly written, with sufficient background material in most cases. The figure is good.
Please describe more fully what are pre- and post-zygotic isolation (lines 43-49), so that those of us who are not in evolutionary biology can understand.
The term 'epigenetic' is currently used more often to refer to a specific set of molecular factors (methylation, histone modifications, chromatin structure, etc.) around DNA that can regulate gene expression. This manuscript's use of the term epigenetic is consistent with its historical use, as developed by Waddington. I find this acceptable.

Experimental design

The experiments are well designed and controlled, and address the stated hypothesis. Methods are adequate, and well described.

Validity of the findings

The results are believable. Since previous studies were very interesting and somewhat surprising, this was a good choice for studies to replicate. The speculations and caveats in the discussion were thorough and appropriate.

Reviewer 2 ·

Basic reporting

Najarro et al report a partial replication of a study by Rosenberg and colleagues from 2010 using two different paradigms testing assortative mating in inbred (CS) and outbred (AllRal) populations of D. melanogaster. While the findings are interesting, some data would be interesting if presented. In addition, while not necessary, a couple of additional experiments would greatly improve this manuscript and the strength of the conclusions drawn.

While the introduction presents the question and hypothesis adequately, it lacks background on previous studies on diet induced assortative mating. Given that it is the main theme of this manuscript, the authors should elaborate more. Specifically, and since the authors chose to replicate Sharon et al 2010, designed after Dodd (1989), the latter should be discussed (especially since it was conducted using wild populations of D. pseudoobscura).

The data presented in the manuscript is interesting though lacking. Figure 1 does not have any informative quality over what’s reported. Specifically, it would be of great interest to present additional data on these experiment: mating success (i.e., number of 3/4-way mating tests performed and total number of matings observed), mating distribution (even though these can be calculated from the supplementary data it is of interest, at least for data involving CS, since there are some discrepancies between the starch choice in 4-way and 3 way mating tests
That are interesting and should be elaborated). A table specifying the different mating observations (SxS, SxC, CxS, CxC) with accompanying statistics would be of interest to readers.

In addition, it is unclear why the authors performed two experiments (one with 3 steps looking at the early effects of transition to starch, and another after 20 generations), but never reported the data separately. If no differences were observed, the authors should report it.

Experimental design

The experimental design the authors used is appropriate to test the hypotheses. The addition of 3-way male/female-choice mating tests is of interest. It should, however, be noted that the time of day in which mating tests are performed is very important and may influence observations. The authors’ impressive observation report states quite a few tests that were performed later in the day.

In addition, the authors’ controls for wing clipping, and other factors that may influence results should be noted. It is an important and necessary analysis for data of this kind.

The authors make no claim regarding the microbiome’s role in these observations. However, it would be interesting to cure flies of bacteria (with abx) either before or after the transition to the new medium. This might also shed light on the role of bacteria in adaptation to starch.

Validity of the findings

The authors’ statistical analysis is sound and appropriate. The authors state their conclusions clearly. The only finding that should be further discussed is the discrepancy between the data for starch choice in 4 way and 3 way tests. specifically, the authors should consider these data, in addition to mating success data, when discussing their suggested mechanism (choice based on individual health). if that were the case, wouldn't there be a bias against starch mates for starch in mating tests?

Additional comments

The manuscript is very interesting and important to substantiate observations on the effects of epigenetic factors on mating preferences and their role in speciation. The suggestion for additional experiments is definitely not necessary, though would be interesting to add to this manuscript, in case you have these data.

·

Basic reporting

The manuscript by Najarro et al is an interesting follow up study to previous work, which found a link between parental diet, micobiome diversity and assortative mating. The design is sound and the results are thought provoking about the importance of non-genetic inheritance in evolutionary processes. However, the study should be couched in terms of what is actually being measured. In addition, the paper is light on conceptual framework which is surprising considering that the ideas about non-genetic forms of inheritance have been around, and studied in flies specifically for a long time, including by the guy who coined the term ‘epigenetics’ (Waddington). Not to mention, there is a lot of recent interest in understanding the importance of non-genetic inheritance. Following are mostly suggestions to improve the breadth of the audience that the study can reach.

Line 64 this use of the term “epigenetically” is unhelpful. Given all the representations of the term “epigenetic”, this requires a definition at the least, but better to keep it to the term non-genetic inheritance (including and especially in the title), which is more accurate. Inheritance of the microbiome might be a good addition to thoughts on non-genetic inheritance more generally- Day and Bonduriansky 2011 Am Nat modeled several different types of non-genetic inheritance, and Jablonka and Lamb consider different forms of non-genetic inheritance (epigenetic is just one), see also figure 1 from Jablonka and Raz 2009 QRB which mentions how symbionts might induce soma to soma transmission of epigenetic variants. The manuscript is missing an actual discussion of epigenetic mechanisms, ie DNA methylation, histone modification, small RNAs. In the Sharon 2010 study the microbiome could be causing these types of actual epigenetic mechanisms, but presumably they weren’t measured. Nor were they measured in this study. In fact more generally in this study, it is a bit of a leap to assume that the differences found were due to the microbiome since this wasn’t quantified. These important details should be reflected in the description of what was found instead of attributing it to the microbiome. In essence, the current study finds only evidence for non-genetic inheritance with intriguing differences between the strains. It could be epigenetically mediated or microbiome mediated or the microbiome changes mediate epigenetic etc. The suggestion of GXE as discussed in the manuscript is still particularly relevant in this context.

Line 106-107 There should be some further discussion of the design in the context of how many generations one would expect in order to see this type of non-genetic inheritance. It took Waddington 14 generations to get the “cross-veinless” phenotype to be inherited without the heat shock (1953 Evolution). Is there something different about behavioral traits and the interaction with the microbiome? Crews et al 2007 is cited, but in a superficial way. Discussing the differences between their results and these could be interesting especially given the parallels with difference in preferences between treated and control animals. Further, might these effects kick in in the outbred lines after more generations sensu Waddington 1953? Might the response in the microbiome mediate a more rapid response for these traits involved in behavior?

Minor comments:
Line 59 The term “ecological adaptation” should really be “genetically based adaptation”. It seems the distinction you want to make is sequence-based differentiation compared to non-genetic mechanisms. I see from searching that this term has been used a bit, but it is unfortunate. To reach a broader audience I suggest the more classic term or at least defining what you mean.

Line 60 the phrase “the environment was having a more direct effect on mating preferences” isn’t clear, especially in the context of the study: 1) its not clear why only 2 generations matters and 2) its not clear what you mean by “the environment” ie that the parent’s nutritional status was having an effect.

Line 140 this description of the “4-way mating” scheme is confusing (perhaps only for non-drosophila, non assortative mating researchers). I take it there were actually 4 flies in each well, but on first read I thought it was just one male loaded into each well, filling wells first with the CWY males and then continuing with the starch males.

line 180 Similarly the wording of the results is not immediately intuitive. On first read the CI is not overlapping with 0 so seems significant, but the random expectation is actually 0.5 not based on overlap with 0.

Line 205 “non-significantly disassortative mate choice” & “non-significantly assortative male-choice” should just be “not significantly different from random mating” in both cases.

Experimental design

The design is sound and the results are thought provoking about the importance of non-genetic inheritance in evolutionary processes.

Validity of the findings

The study should be couched in terms of what is actually being measured as mentioned above. With specific reference to the microbiome and what was measured in the study, the term "epigenetic" is not appropriate and should be changed to "non-genetic".

---

## Round 0.2 · accepted · Accept

Your responses to all reviewer criticisms and requests was complete and appropriate. Thank you for your prompt turnaround, and for this nice addition to the literature.